# A Review of Gas Measurement Set-Ups

**DOI:** 10.3390/s22072557

**Published:** 2022-03-27

**Authors:** Łukasz Fuśnik, Bartłomiej Szafraniak, Anna Paleczek, Dominik Grochala, Artur Rydosz

**Affiliations:** Institute of Electronics, AGH University of Science and Technology, Al. Mickiewicza 30, 30-054 Kraków, Poland; lfusnik@agh.edu.pl (Ł.F.); paleczek@agh.edu.pl (A.P.); grochala@agh.edu.pl (D.G.); rydosz@agh.edu.pl (A.R.)

**Keywords:** gas sensors, measurement set-up, static measurements, dynamic characteristics, in situ, operando

## Abstract

Measurements of the properties of gas-sensitive materials are a subject of constant research, including continuous developments and improvements of measurement methods and, consequently, measurement set-ups. Preparation of the test set-up is a key aspect of research, and it has a significant impact on the tested sensor. This paper aims to review the current state of the art in the field of gas-sensing measurement and provide overall conclusions of how the different set-ups impact the obtained results.

## 1. Introduction

The history of modern gas detection dates back to the nineteenth century [1]—here, a short history is provided for a better perspective of how gas-sensing measurement has developed over the years. Briefly, gas detection was initially limited to safety applications. People working in mines from ancient times were aware of the need for ventilation. In coal mines, the first detector of methane or a lack of oxygen was a torch which, in the absence of oxygen, would go out and would ignite in the presence of methane. Unfortunately, such a working principle carried a lot of risks [2,3]. In the nineteenth and early twentieth centuries, canaries were brought into the tunnels as an early gas detector. The canaries would stop singing in the presence of methane, carbon monoxide, and carbon dioxide, and with a rapid increase in gas concentration in the atmosphere (leakage), it would go into convulsions and die unless it was quickly removed from the harmful atmosphere. It was a clear signal for miners to flee such an area [2,3].

In the industrial era, the first gas detector was the flame safety lamp (or Davy lamp), invented by Sir Humphry Davy in 1815. It was used to detect the presence of methane in underground coal mines. The idea behind the operation of this lamp was very simple; it burned with an oil flame set (calibrated) to a certain height in the open air. In order to prevent the lamp from exploding or starting a fire, the lamp flame was placed in a glass sleeve with a mesh flame arrester. The height of the flame varied depending on the composition of the atmosphere around the lamp. The flame was higher in the presence of methane and lowered in the absence of oxygen. Interestingly, in some parts of the world, such a solution is still used today [4,5,6,7]. In 1928, Vertucci patented a device for detecting the presence of combustible gases in air in the US patent office, which reacts to the presence of flammable gases by increasing the temperature of the filament caused by oxidation on its surface [8].

The gas-sensitive properties of semiconductor materials were discovered in the 1920s [1]. However, the first idea of chemical-resistive gas sensors was born in 1952, when Brattain and Bardeen were the first to describe the gas-sensitive properties of germanium [9]. The history of chemoresistive gas sensors from their beginning is described in detail by Neri [10]. One of the next important steps in the field of gas detection was the discovery of changes in the electrical conductivity of zinc oxide (ZnO) in 1955 [11]. The first commercially available gas sensor was the TGS (Taguchi Gas Sensor from Figaro Engineering Inc.), which was launched on the market for methane detection TGS109 [12] and was based on tin dioxide [13,14,15]. In 2016, Mizsei briefly summarised the history of the development of semiconductor gas sensors. This summary included statistics on scientific publications in this field, trends, a literature review, equipment and technology, and a theory of operation [16]. 

Regardless of the target gas detection, the gas measurement stands are crucial elements for gas sensors development. In 1995, Endres et al. presented the construction and description of a set-up constructed for measuring sensor properties [17]. The authors described the structure of individual subsystems, including gas dosing, temperature and humidity control, measurements, and control and communication between the individual components of the set-up. The authors supported their design of the test set-up with promising test results of gas sensors [17]. They pointed to the growing importance of the measurement system, which, according to the authors, should translate to the researchers paying more attention to this area. This especially applies to lowering the dosed gas concentrations while maintaining an appropriate level of accuracy, increasing the number of mixed gases simultaneously in one measurement, the repeatability of measurement cycles, and the effectiveness of measurement methods.

In 2017, Singh et al. presented a review of wide-range gas sensors, taking into account the influence of the conditions in which sensors work on gas detection. The presented results include, among other parameters, temperature, humidity, flow, pressure, conductivity, and heating methods [18]. A few scientific papers have focused on measuring set-ups for testing gas sensors. Gracheva et al. [19] and Kneer et al. [20] have presented works on the construction of measuring set-ups for measuring the response of sensors to the presence of gas, mostly based on the resistance changes. By contrast, Smulko described a test set-up for measurements using two methods at the same time (measuring the fluctuation of resistance and the base voltage of the sensor) [21].

In 2019, Nasiri et al. analysed the possibilities of developing the use of gas sensors in diagnosing human health based on the analysis of exhaled air. Their considerations are directed towards nanotechnology and preparing a review of work on the use of nanosensors in medicine, their production, application, and possible directions of development in this branch of gas detection [22].

Over time, the cost of production of gas sensors has decreased significantly with a simultaneous increase in their efficiency. In recent years, gas-detection technology has grown in importance due to its availability and widespread use in the following areas: air-quality monitoring [10,23,24,25,26,27,28,29,30,31], industrial production [24,25,31,32], food testing [10,21,26,27,33,34,35], automotive [10,23,27,35,36,37], industry [24,25,35,38], medical applications [10,22,25,26,27,35,39,40,41], fire warning [25], hazard monitoring [25,27,31,35,42,43,44,45], monitoring of gases at ambient conditions [46,47], that is the one of the most actual important approach, etc. There have even been extensive reviews of the use of gas-sensor systems, including the work of Hsieh and Yao [26], and in 2019, Feng S. et al. published an interesting review of sensor technology with an emphasis on smart technology, including the Internet of Things (IoT). It contains a detailed division of sensors, areas, and examples of their use. Part of the article is also devoted to the processing and interpretation of data obtained from gas sensors [27]. The global gas-sensor market, according to the literature [48,49], was valued at 2.19 billion USD in 2019 and is projected to grow between 2020 and 2027, as expressed by a change in the compound annual growth rate (CAGR) from 6.3% to 8.3%. 

The growing demand for gas sensors leads to the increasing interest of scientists in this field. MOS (metal oxide semiconductor) sensors are the most popular gas-detection materials mainly due to their low cost and high sensitivity. In scientific research, apart from the leading actor—in this case, the gas sensor—the supporting actor (the measuring set-up) is also very important. Despite the fact that there are many publications on gas sensors, there is no review on measurement methods and measuring set-ups, which makes it impossible to directly compare the obtained results. Usually, in the papers describing the tests of the properties of gas sensors, the test set-up is only briefly described [50,51,52] or often not at all [53,54,55]. Therefore, it is reasonable to prepare an overview of gas-sensor measuring set-ups and to try to characterise their properties and parameters.

Regardless of the fabrication of the set-ups, the gas-sensing characteristics result mainly from the material properties between gas-sensing layers and target gas molecules. It is essential to take this into account, especially when a new set-up will be designed. The theoretical role in the exploration of gas measurements cannot be neglected, and it should be a subject of a separate review. However, some crucial information could be found in the recently published papers by D’Olimpio et al. [56], Li, J.-H. et al. [57], and Pineda-Reyes, A.M. et al. [58].

The aim of this paper is to make it possible to compare the results obtained from measurements of gas sensors, which are based on various research techniques and various existing measurement systems. An analysis of the test set-ups used over the last 50 years has been carried out, but the technical possibilities now and then need to be considered. The measuring set-ups for testing resistive-type gas sensors are presented and discussed. It has to be underlined that several various types of gas sensors are currently investigated [59], such as electrochemical, catalytic bead (pellistor), photoacoustic, optical [60], and semiconductor gas sensors. The breakdown of gas detection methods is presented in the form of a diagram in Figure 1 [59]; however, metal-oxide-based sensors are widely used in industrial and consumer applications. An overview of the advantages and disadvantages of each gas detection method is provided in Appendix A. 

## 2. Semiconductor Gas Sensors—Electrical Resistance Measurements

The measured output signal of semiconductor gas sensors is resistance and/or capacitance, and more precisely, its changes in the presence of gas. However, the resistance of a semiconductor gas sensor is significantly dependent on many other factors apart from the presence of gas. These factors include the operating temperature, relative humidity, and the overall composition of the tested atmosphere. Figure 2 shows the main parameters that must be controlled in the stand for measuring the sensor response of semiconductor gas sensors. Information about the sensor’s response to gas without precisely defining the conditions for carrying out the measurements is useless. The measuring set-up, apart from the function of measuring the sensor’s response to gas, should enable the determination of the conditions of its optimal operation, as well as their stable maintenance during the entire measurement process.

Based on the literature review focusing on measuring set-ups, it can be seen that the basic guidelines for constructing set-ups for testing the response of gas sensors do not change over the years. The measuring set-ups consist of basic (necessary) elements, such as: gas sources, gas lines, flow meters, measuring chambers, devices measuring the sensor response, control systems, and equipment for gas neutralisation (the safe discharge of gases). Additional elements of the measuring set-ups that significantly improve their accuracy and multiply their measuring capabilities are, among others: devices stabilising the measurement conditions in the measuring chamber, i.e., systems controlling the relative humidity and temperature of the gas mixture in the measuring chamber, as well as the flow and concentrations of individual gases. Measurement set-ups are increasingly evolving towards performing several measurements of a simultaneously working sensor for both in situ and operando measurement methods. 

### 2.1. Gas-Sensing Measurement Set-Up Parameters

The aspects of occupational health and safety in laboratories are widely described in many publications—examples can be found in the literature [44,45,61]. Therefore, the gas-sensing measurements are conducted under exposure to various gases, depending on the dedicated purpose. The test sample is usually placed in a measuring chamber connected to a sealed gas installation. Often, harmful or even toxic gases are used for measurements. Various gases are connected to the chamber through installations; for safety reasons, the room with the station must be equipped with efficient ventilation and a system for monitoring the concentration of gases in the atmosphere.

#### 2.1.1. Discharge and Neutralisation of Gases

The gases flowing through the measuring chamber with the sensor must be safely discharged or neutralised. Most often, measurements of the sensor properties of materials are performed in dedicated laboratories (inside buildings). Such rooms must meet a number of stringent requirements of occupational health and safety regulations. Gases must be stored and delivered under safe conditions [61]. Firstly, gas installations must be tightly sealed, and laboratory rooms need to be well ventilated. A significant problem after supplying the gas or gas mixture to the chamber with the tested sensor is the removal of the gas from the chamber (a vacuum pump is often used for this [62]) and its neutralisation or disposal. This aspect of the measuring set-up does not affect the results of sensor measurements, so it is often omitted in publications or marked in a very general way in the schemes of measuring set-ups if it is included in the paper [34]. One of the common solutions ensuring the health and safety of the station’s operation is carrying out measurements that detect gas under a hood [36]. Another popular and safe method of neutralising gas mixtures used for sensor measurements is the neutralisation of hazardous gases in a bubbler filled with a solution that reacts with the gases used [34,63]. Such a solution has been used by, among others, Barauskas et al. [64] using 0.5 mol NaOH to neutralise SO_2_ and CO_2_.

A similar solution of gas neutralisation in sensor measurements was used by Adeniyi et al. They used hydrogen sulphide in an experiment, which was first released into a dry gas trap after each measurement, from which it travelled to a scrubber with a 20% KOH solution (Figure 3), and then on to a second scrubber with the same solution and finally to another dry gas trap. After the measurement process was completed, the entire measurement system was purged with clean nitrogen [45]. This simple solution is very effective because the output of the system is pure gas. Hazardous ingredients are neutralised in gas traps (Figure 4). Another popular solution is the use of a burner to burn the flammable gas [65]. Some articles only provide brief information on gas neutralisation, e.g., with the use of adsorbers (Figure 4 and Figure 5) [63,66]. By contrast, in the case of using a sensor-measuring system outside, when testing the presence of gases in the atmosphere (surroundings), it is enough to use a fan to force the gases to move within the measuring chamber (in the vicinity of the sensor). Such a solution was proposed by Das et al. in a publication about a mobile robot for the detection of hazardous gases [67]. The problem of removing and utilising gases after the measurement is significantly simplified if gases that are inert with regard to health and the environment are tested, e.g., those that normally occur in the environment. Then only the question of removing gases from the measuring chamber remains to be solved. When measuring small-volume measuring chambers and small amounts of gases dispensed into the chamber (e.g., by means of a syringe), gas is removed from the measuring chamber by opening the chamber and releasing the contents to the environment [68]. 

#### 2.1.2. Materials Used for the Construction of Measuring Set-Up

When testing gas sensors, toxic, poisonous, and highly reactive gases are usually used, and it is important to eliminate as many interferences as possible that may occur during the measurements. One of these is the reaction of the measuring set-up elements with the gases flowing. The effects of such reactions and contamination in the gases or air used can introduce huge measurement errors or even give false results. For this reason, materials that do not react with oxidizing and reducing gases are used to make the elements of the sensor test set-ups. For example, measuring cells are made of glass (e.g., borosilicate glass [51]), acrylic glass [69], stainless steel [17,20,62,70], aluminium, Teflon [63] and various other plastics [66]. These are non-reactive materials, and additionally, they are resistant to high temperatures that can change over a wide range. Kamiko et al. used a stainless steel chamber coated inside with Teflon to avoid gas adsorption [71].

#### 2.1.3. Test Chamber, Gas Lines, and Connections

Generally, the gas flow, including the target gas and ambient gas (mostly synthetic air, argon, or nitrogen), is regulated by the utilisation of mass flow controllers (MFCs), which also allows researchers to dilute the testing gas and measure the gas sensor’s response under exposure to various concentrations of the target gas. Thus, the various concentration ranges are the obvious parameter that influences the response of the sensor. In sensor test set-ups, several mass flow meters are most often used. The set-ups are prepared for the use of several gases in a shift [20,72,73,74,75]. This is a better solution (measurement errors were limited) than using one gas line and one flow meter with simultaneous cylinder replacement or switching [76], although this is much more expensive. 

#### 2.1.4. Rate of Gas Flow

When talking about flow meters and gas connections, the role of gas flow in the measuring chamber cannot be ignored. Gas flow is also of great importance during measurements. The flow rate strongly influences other parameters important during the measurement procedure, such as temperature and humidity and their stabilisation [17,69]. The higher the flow, the more difficult it is to stabilise other parameters. The set-up requires, for example, additional preliminary stabilisation of the gas temperature. The humidifying system must also be more efficient. The total flow can also be used to regulate the concentration of the detected gas to an appropriate extent (dilution, e.g., in air, nitrogen, or argon). A higher flow also means less delay in the system; the gas moves faster through the station. The delays in the measuring system are also strongly correlated with the volume of the gas lines and the measuring chamber—the larger the volume, the more time it takes to replace it. Examples of the exact values of gas flow through the measuring set-ups or their ranges are to be found in the literature [30,51,64,72,77,78,79,80,81]. Kneer et al. even provide the exact time of stabilisation of the conditions in the chamber after changing the gas concentration [20].

The gases concentration is regulated in several ways in the sensor-measurement set-up. The simplest solution to the lack of regulation is feeding gas from a cylinder with a specific concentration. The system of mass flow meters makes it possible to regulate the gas concentration within a certain range [79]. The total flow is predetermined and does not change during the measurements [79]. When there is a need to use flows in a measuring chamber with a wide range, mass flow meters duplicated in each gas line are often used. An interesting example of the miniaturisation of a gas-detection system is the proposal presented by Muller et al. [81], in which the chamber with gas sensors has dimensions in the order of single millimetres. The kit also included a micropump and an ozone generator. To test this type of solution, test stands that will reflect the working conditions of the tested element are needed.

#### 2.1.5. Volume of Test Chamber

Volume is also one of the most important parameters of the measuring chamber. In general, the smaller the size, the faster the entire gas volume is exchanged. The processes taking place in the chamber are more dynamic. The reaction of the test sensor to gas is faster, i.e., the measurement error caused by delays in gas flow through the volumes of connections. For example, Harvey et al. described the development of a semi-automatic measuring chamber for simultaneous characterisation tests (Figure 4a) [69], where small volume was used. Endres et al. [17] used two chambers in their test set-up; these were used interchangeably depending on their specific needs. One was linear and smaller, while the other had a greater volume and a cylindrical shape. In their paper, the researchers emphasised that when designing and building a new measuring chamber, they try to keep it as small as possible [17]. An example of a simple, functional set-up is presented by Bum-Joon et al. [72], in which the measuring chamber has a volume of 2000 cm^3^, and the gas flow through the set-up is 2000 sccm. Within one minute, the entire volume of the chamber is replaced. Järvinen et al. [77] presented a measuring set-up with a chamber with a volume of 1500 cm^3^ and flow regulation in the range of 1–1000 sccm. Kneer et al. made and described a sensor set-up with a measuring chamber with a capacity of 255 cm^3^ and a gas flow through the stand of 1000 sccm. Researchers declare that with such parameters, after changing the gas concentration, the conditions in the chamber reach a steady state after about 46 seconds [20]. Casanova-Cháferairtight et al., in their research, used a measuring chamber with a volume of 35 cm^3^ and with a total gas flow through the station at the level of 100 sccm [82]. Sberveglieri et al. [78] used a measuring chamber with a volume of 1000 cm^3^ in the constructed set-up. With appropriate manipulation of the volume of the measuring chamber and the gas flow through the station, the dynamics of gas exchange in the measuring chamber can be controlled across a very wide range, which will be discussed in detail in the chapter on static and dynamic measurements. Generally, in order to reduce inertia in the measuring stand of gas sensors, measuring chambers with as small a volume as possible are used. Such a solution has been presented by Huber et al. [83]. Photographic documentation of the prepared measuring chamber with a volume of 20 cm^3^ is presented in Figure 4c. All elements, including the tested sensor, are mounted on both sides of the printed circuit board (including temperature and pressure sensors). In order to prevent self-heating of the electronic components, no active electronic components are mounted on the board. All signals are routed externally through appropriate connectors. The described gas-measuring chamber enables measurements up to pressures of at least 10 bar [83]. An additional advantage of using small-volume chambers and the shortest possible connections is also gas and energy savings, and thus the measuring station is more environmentally friendly. There is also an example of a test bench with a large-volume chamber in which a 100 dm^3^ chamber is used [64]. The large measuring chamber allows multiple sensors, in this case, 50, to be placed at once. Drawings and photos of various measuring chambers and other elements of test stands for gas-sensitive elements can be found in Figure 3, Figure 4, Figure 5, Figure 6 and Figure 7.

**Figure 4 sensors-22-02557-f004:**
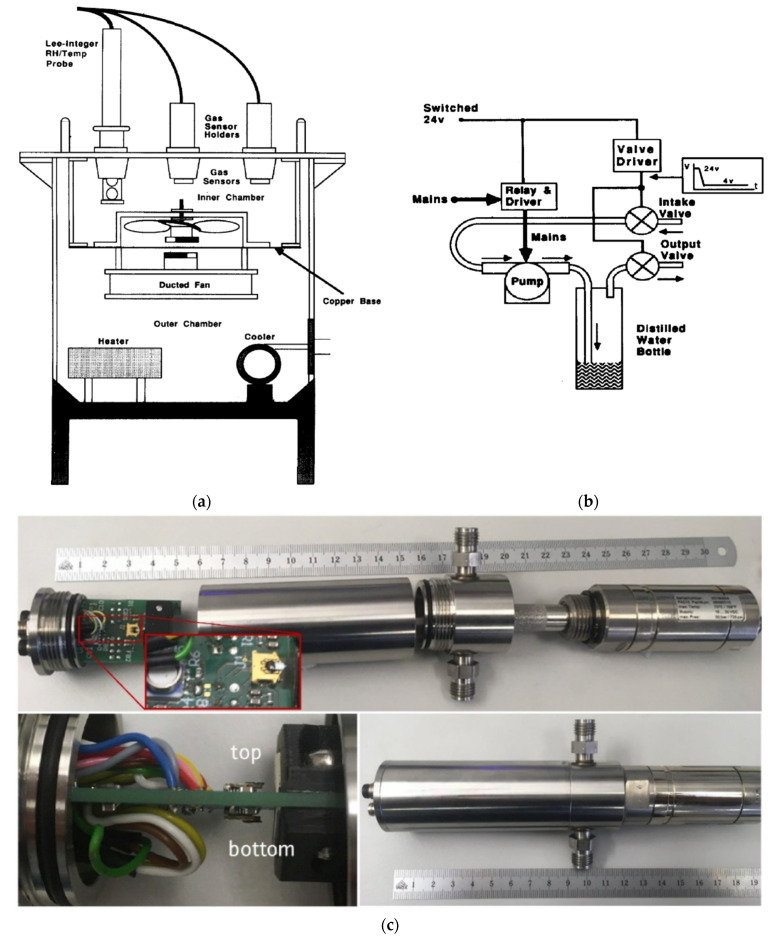
(**a**) Experimental chamber, not showing the humidifying circuit. Reprinted with permission from [69]; (**b**) The humidification circuit. Reprinted with permission from [69]; (**c**) Top: General view of the measuring chamber with sensor printed circuit board (PCB), gas cylindrical chamber with inner diameter of 30 mm, fluidic connections and dew point sensor. Zoom: top view of the sensor PCB with the pressure and temperature sensor and the first microcantilever in front of a permanent magnet. Bottom left: the second cantilever is placed on the rear side of the PCB. Actuation occurs by supplying a small alternating current (AC) intensity over the metal coil on the cantilever tip. Bottom right: sensor PCB and dew point sensor mounted in a pressure-tight measuring chamber (numbers in the scale correspond to cm). Reprinted with permission from [83].

Interestingly, a sensor-test set-up with only one gas line was presented by Sun et al. in an uncomplicated mini-system for gas detection—environmental monitoring [30]. Or an even more minimalist solution was shown by Afzal et al. built an installation without any flow meters or gas lines [68]. A syringe was used to feed gas through a special partition to the measuring chamber, and a plug was employed for ventilation. There was no steady flow in the sensor chamber. The advantage of such a system was undoubtedly its simplicity and low costs, including its low gas consumption, which in turn resulted in no need to build a drainage system and a method for the utilisation of gases used in the measurements. A round aluminium disc with a 32 m diameter and 11 mm thickness with an integrated pencil heater was used as a heating element in the chamber (Figure 5a). Heated elements are thermally insulated from the floor through a Teflon spacer. This solution was used to heat the local film. An Alumol Chromel thermocouple was used for temperature measurements. The tested sensor was mounted above the heating element. The cable connections for resistance measurements are spring-loaded. All this was placed in a glass cylinder with a diameter of 110 mm and a length of 115 mm, ending on each side with aluminium discs. A flange coupling with an O-ring together with round aluminium discs was used as a seal for the chamber closure. This solution ensured the air-tightness of the chamber. All electrical connections were threaded through the bottom plate. A digital multimeter was used as a resistance meter (Figure 5a,b) [68].

#### 2.1.6. Mix Chamber

If it is necessary to mix gases in the test set-up (in cases where a ready mix from the cylinder was not used), an additional mixing chamber should often be used in which the supplied gases have enough time and volume to mix evenly [34,65,69,71]. An example of a measuring chamber with a fan fitted to ensure good gas circulation is shown in [69]. Mixing gases in the gas lines themselves without a mixing chamber may prove to be ineffective, contrary to appearances, in such a small volume and shape with specific geometric dimensions (small diameter and relatively large length of gas line) the gases may not mix properly, which may be a reason for receiving incorrect measurement results.

**Figure 5 sensors-22-02557-f005:**
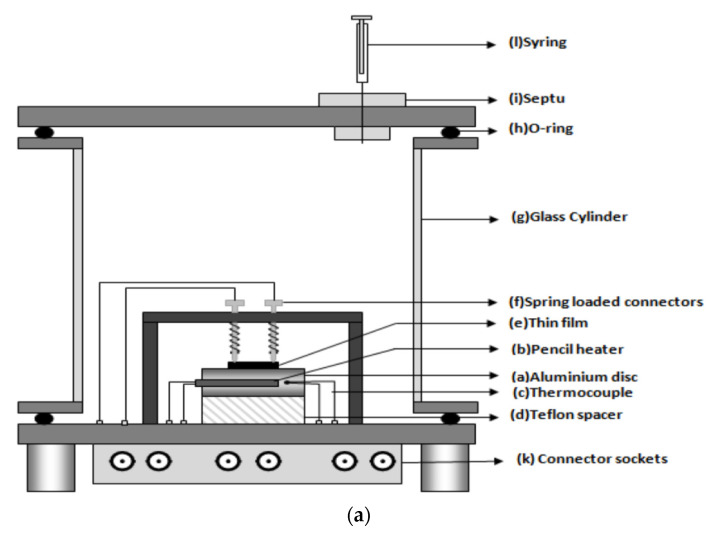
(**a**) Schematic diagram of proposed gas-sensor unit. Reprinted from [68] under CC BY-NC licence; (**b**) Photograph of the proposed gas-sensor set-up. Reprinted from [68] under CC BY-NC licence.

#### 2.1.7. Temperature

Semiconductor-based gas sensors operate at higher temperatures [23]; this is related to the theory of semiconductors, which shows that the study of gas sensors based on this technology requires the adjustment of the sample (tested sensor) temperature across a wide temperature range. Semiconductor gas sensors usually work optimally at a temperature of several hundred degrees Celsius (most often, it is in the range of about 200 to 400 °C). Research works are aimed at inventing a sensor that would work stably, quickly, and reliably at room temperature, or that its optimal working temperature should be as close to room temperature as possible [82]. This solution simplifies the design of the device and its use and is also economical and ecological.

It follows that when working on new gas sensors, it is useful in the set-up to be able to regulate and stabilise the sample’s working temperature during the tests (such tests are often quite long, lasting even weeks or months). There are many solutions to this problem. One of these is placing a gas-sensitive layer of resistance heaters on the previously prepared electrodes. In such a solution, the electric current was fed to the heaters placed in the substrate, which, flowing through the resistance heaters, heats the sample [20,84]. Examples of the geometries of micro-heaters used in sensor substrates are shown in the literature [84] and in Figure 6 [83]. Simons et al. [70,79] used the source meter sensor for heating, which powered the previously described heaters in the substrate. A spectral pyrometer was used to control the temperature level. The great advantage of this method of measuring the operating temperature of the sensor is that it is a wireless method. This solution allows maintaining the preset operating temperature of the tested sensor while maintaining the room temperature (or other temperature depending on the temperature requirements) of the gases supplied to the sample.

**Figure 6 sensors-22-02557-f006:**
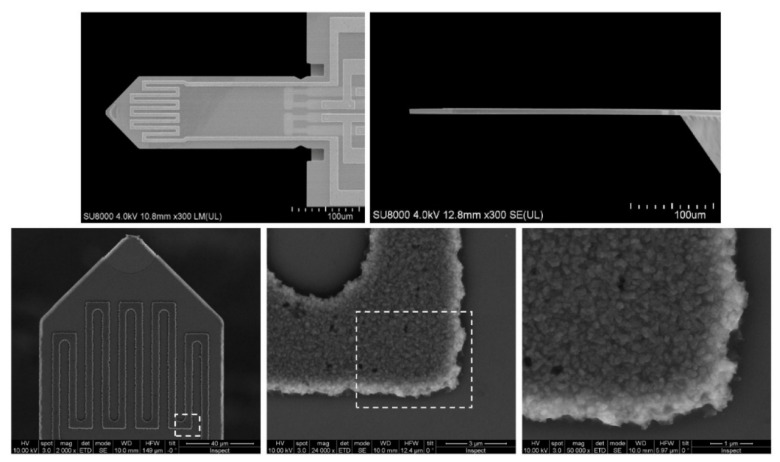
Top: top and side view of the silicon microcantilevers from SCL-Sensor. Tech. (PRSA-L300-F50-TL-PCB) [85] used in this study. The cantilevers have a length of 300 µm a width of 110 µm and a thickness of between 2.5 to 4 µm. Bottom: Scanning electron microscope (SEM) images of the cantilever top surface and details of the heater coil. Reprinted with permission from [83].

A commonly used tool for measuring the temperature in the measuring chamber is a thermocouple [17,78]—it is characterised by high accuracy and resistance to high temperatures. The disadvantage of this solution is the necessity to lead electrical connections from the chamber (wire measurement).

Another solution is to heat the table on which the tested sensor is placed or the entire measuring chamber [69]. In addition to heating the sample to the set temperature, this solution also causes heating of the gas that is introduced into the chamber. It is rare for the sensor to work with gases at a temperature of several hundred degrees (not counting sensors in internal combustion engines and other solutions where such operating conditions are normal, of course). In this case, an adequate gas flow must be ensured through the measuring chamber, large enough that the gases do not have time to heat up before they reach the sample. It is quite a complicated system, and often, there is also the problem of stabilising the temperature of gases reaching the measuring chamber. Gases from cylinders, when fed to the station, usually expand (the pressure in the test station is kept lower than in cylinders or the supply system), which is associated with lowering the gas temperature. In the case of small volumes of the measuring chamber and gas lines, gas with a temperature lower than room temperature or another set temperature reaches the measuring chamber. The gas does not warm up on its way from the cylinder to the sample. This lower temperature, even by a few degrees Celsius, can have a huge impact on the reaction of the sensor (semiconductor materials are sensitive to temperature changes as their conductivity strongly depends on temperature). Finally, it may turn out that the impact of changes in the gas temperature, and not the gas itself, on changes in the resistance of the sample was examined. For this reason, in measuring stations, it is possible to heat/stabilise the temperature of the gases supplied to the measuring chamber in order to minimise temperature fluctuations in the measuring chamber. Such solutions involve heating a section of the gas line leading to the measurement chamber or stabilising the temperature of gases in the mixing chamber. The use of the mixing chamber has a positive effect on reducing temperature fluctuations of gases going to the measuring chamber.

#### 2.1.8. Humidity

Relative humidity has already been mentioned several times before and is another very important parameter in the sensor measurements of semiconductor gas sensors. When testing a new sensor, it is necessary to determine the influence of relative humidity on sensor operation. If the sensor is dedicated to working in strictly defined conditions—i.e., where, for example, the relative air humidity in question is constant or changes only across a small range—it is enough to check its effects in the range of interest. The surest way to control the humidity level is to place the humidity sensor in the measuring chamber, and then you know precisely what humidity the tested sensor is working at [69,82].

#### 2.1.9. Pressure

Another important factor when measuring sensor properties is the pressure in the measuring chamber. Huber et al. [83] proposed a solution, and the scheme is presented in [83]. The pressure in the measuring chamber was regulated by a special valve in the range of 1 to 10 bar, which extends the possibilities for the operation of the device in various industrial installations. The temperature of the measuring chamber was controlled by a jacket rinsed with water from a thermostatic bath. The measuring chamber is also equipped with a humidity control system. It is worth emphasising that a portable device is used as a control unit instead of a desktop computer [83]. Another example of the use of pressure in sensor measurements was the measurement procedure described by Zhang et al. [62]. After determining the temperature, the air was pumped out of the chamber to create a low vacuum at a gas pressure of 5 kPa; the measured gases and the ambient gas N_2_ were then introduced in a controlled manner until the pressure in the chamber increased to 1 atm.

### 2.2. Examples of Gas-Sensing Set-Ups

An example of an automated measurement system with the possibility of using several gases at the same time is shown in Figure 7—temperature, humidity and flow stabilisation is included. Figure 7a shows a schematic of a gas detection system where the three gas lines are connected in parallel, and there is also a separate line with a humidifying system. Figure 7b shows a sketch of the possible realisation of the idea of Figure 7a. Figure 7c is another example of the measuring system from Figure 7a, which shows measurements with an extensive system of gas-dosing lines.

For example tests, the block diagram of the measurement procedure is presented in Figure 8d. Figure 8a shows an example of how the various relative humidity (RH) can be obtained. Figure 8b,c show the diagrams of the measuring chambers used interchangeably in one station depending on the properties of the measured sensors. The measurement chambers are used to measure many sensors simultaneously. The linear chamber (shown in Figure 8b) provides a linear gas flow with good ventilation of the entire chamber. The disadvantage of this solution is the linear arrangement of the sensors, through which the gas flows one by one. This solution may lead to the heating of the gas flowing by successive sensors, which may significantly disturb the measurement of sensors located further from the gas inlet. The circular chamber solves this problem; its design allows the simultaneous flow of gas with the same parameters through all sensors on its circumference. This solution, however, is characterised by slower gas exchange (the chamber time constant is greater), which makes it unsuitable for testing faster-reacting gas sensors (with a shorter time constant).

**Figure 7 sensors-22-02557-f007:**
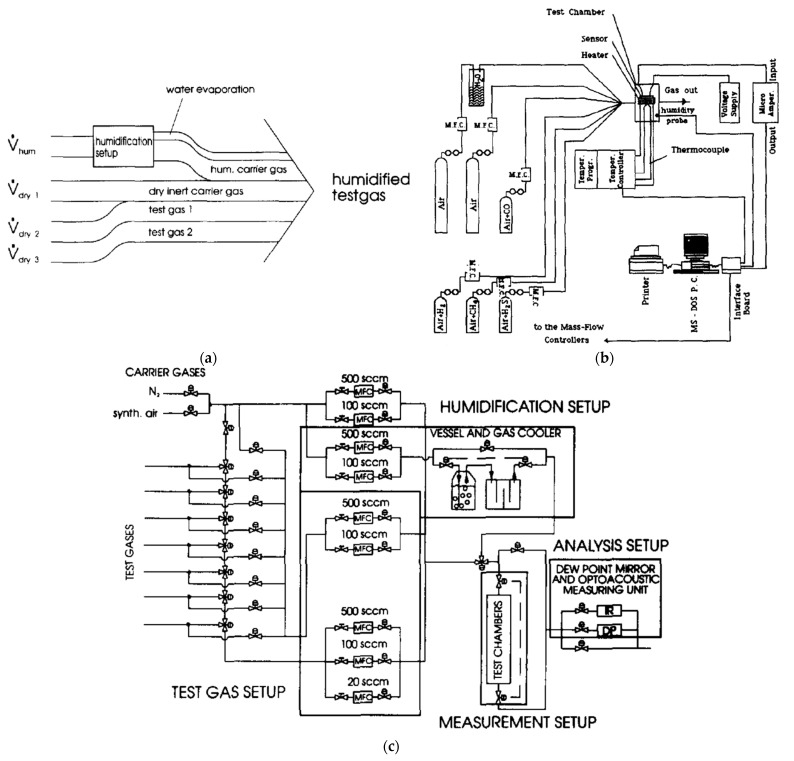
(**a**) Plan of the gas-sensor measurement set-up. Reprinted with permission from [17]; (**b**) Plan of the system measuring the electrical responses of a thin-film gas sensor. Reprinted with permission from [78]; (**c**) Schematic drawing of the volumetric gas-mixing method. Reprinted with permission from [17].

It can be observed that the same sample tested on two different measuring systems will provide two different results, depending on the various parameters. These differences may be either small or high. Figure 9a shows a schematic of a test set-up that allows the researcher to perform measurements on an active odour-detection system. Different gas mixtures are fed alternately to the measuring chamber with the sensor matrix. The composition of one (reference) mixture is known. The second mixture has an unknown composition. The control computer software determines a pattern match between the known and the unknown gas mixture. This measurement is performed iteratively by changing the mixture ratio in accordance with the algorithm of the adaptive control theory—the regulator self-tuning method (STR). The component pairs are mixed by regulation of the MFCs [52].

Figure 9c shows a schematic of a portable electronic nose test set-up. A 3D printing method was used to make the measuring chamber. Nylon was used as a construction material. The device includes: sensors, measuring chamber, pump, electromagnetic directional valves, sample chamber and volatile organic compounds (VOC) absorbers. In this solution, a constant and stable gas flow through the measuring chamber (a necessary condition for repeatable measurements) is guaranteed by the pump, the speed of which is controlled in real-time using the proportional–integral–derivative (PID) positioning algorithm during each measurement. Using a pump instead of mass flow meters is cheaper, which is very important for a mass-produced device. In this case, the flow rate was set to 2000 sccm. Gas paths were switched by means of electromagnetic directional valves. When testing gas sensors, consider the working environment in which various interfering gases, such as volatile organic compounds, may appear. For example, ethanol and formaldehyde are standard, commonly used solvents. They can lead to a significant response of MOX (metal-oxide) sensors. An interesting solution in the described portable set-up is the elimination of the influence of potential gases that interfere with the measurement, i.e., volatile organic compounds. For this purpose, two absorption devices were used to absorb VOCs at the inlet and outlet of the purge gas path. The problem of humidity fluctuations in the measuring chamber was solved by designing a 500 cm^3^ buffer chamber, which significantly reduces humidity fluctuations [66].

Figure 10 shows a diagram of the set-up, sample preparation, gas for testing and sample measurement results. The control of gas dosing and its flow takes place in this case by means of a mass flow meter and a solenoid-controlled valve with a possible range of 0–500 sccm. Nitrogen was used as a carrier gas and also for purging and cleaning the measuring chamber. The measuring chamber was made of Teflon and had a volume of 100 cm^3^. Changes in the frequency of the sensor were measured with a quartz microbalance. The set-up was controlled by an interface developed in the LabVIEW environment. The set-up made it possible to control and stabilise the temperature and humidity in the measuring chamber. The flow through the chamber was 200 sccm [63].

**Figure 9 sensors-22-02557-f009:**
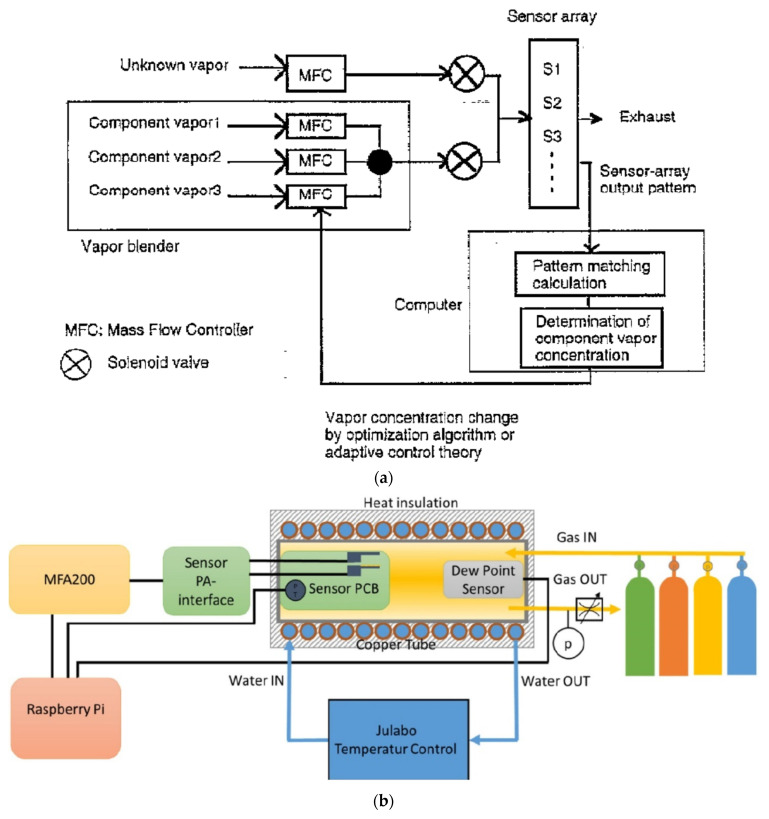
(**a**) Schematic diagram of active sensing system. Reprinted with permission from [52]; (**b**) Schematic representation of the measurement set-up. Reprinted with permission from [83]; (**c**) Schematic view of the portable cigarette odour measuring system. Reprinted with permission from [66].

**Figure 10 sensors-22-02557-f010:**
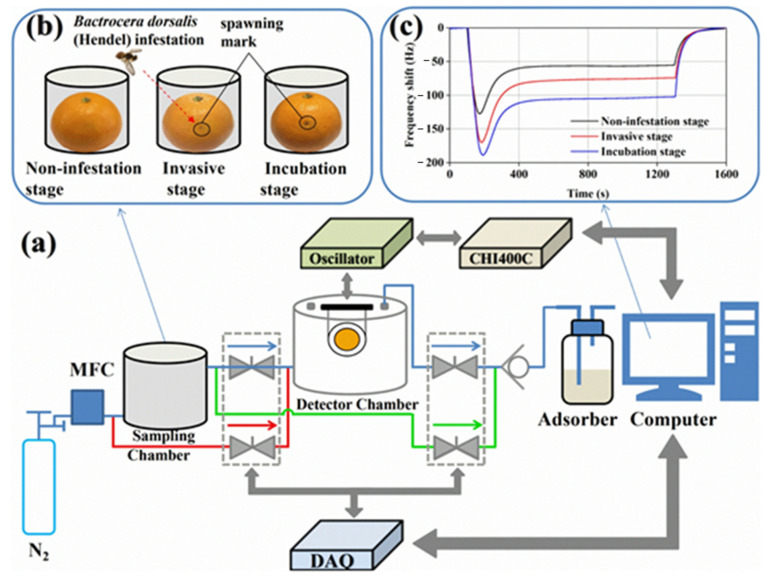
(**a**) The developed gas-sensing system with the EC-coated QCM sensor; (**b**) Preparation of sampling gas; (**c**) The measured frequency changes of the sensor for citrus samples at different times. Reprinted with permission from [63].

### 2.3. Electrical Resistance Measurements

Generally, the semiconductor gas sensors are deposited on the gas sensors substrates made in the form of interdigitated electrodes (IDEs). Therefore, the electrical resistance is measured between two electrodes as schematically presented in Figure 11a. By modelling the tested sensor as an electrical circuit (example in Figure 11a), attempts can be made to predict its behaviour under various conditions. The more accurate the model, the better it reflects the actual behaviour of the sensor. By having a very good model, it is possible to design a sample based on the results of the model. Figure 11b shows a typically measured frequency response for such a sensor system. These data were obtained with the sensor exposed to clean, dry air. On this basis, using a combination of measurement data and data from the literature, values were obtained for different elements in an equivalent electrical circuit (Figure 11a) [85]. Figure 11c shows several resistance characteristics combined for different sensors. The measurement cycle consisted of a 3 min baseline purge (purging gas lane Phase I), a 3 min sampling (acquisition gas lane Phase II) and 3 min baseline recovery (purging gas lane Phase III). First, the air passes through these Phase II) and a 3 min primary recovery (purge gas path Phase III) [66]. In the next Figure 11d, there is also the characteristic of changes in resistance, although not of a single measurement cycle (air/gas + air/air) but many measuring cycles, one after the other, additionally at different temperatures. Thanks to this measurement, it is possible to find the temperature of the optimal sensor operation or the lowest temperature at which the sensor works sufficiently [16]. The electrical resistances are measured under exposure to the target gas and the reference gas (mostly synthetic air), R_gas_ and R_air_, respectively. Both R_air_ and R_gas_ have a significant relationship with the surface reactions taking place [86]. One of the basic parameters characterising the gas sensor is its reaction to the gas that the sensor is to detect. Two terms are most commonly used to describe a sensor’s response to gas—response and sensitivity. The authors of research often use the terms sensitivity and response of a gas sensor interchangeably.

Then, the gas sensor response/sensitivity (*S*) is defined as R_gas_/R_air_ for oxidising gases or R_air_/R_gas_ for reducing gases, respectively.

Usually, the sensor response (*S*) is defined by the formula
(1)$$S=\frac{R_{0}}{R_{g}}\;\mathrm{or}\;S=\frac{R_{g}}{R_{0}}$$
where:$R_{0}$—sensor resistance without the presence of gas,$R_{g}$—sensor resistance in the presence of gas.

Conductance or current is also used instead of resistance. As can be seen from the dependence (1), the sensor response to gas is a dimensionless quantity.

The sensitivity of the sensor, also denoted by (*S*), is commonly defined as follows:(2)$$S=\frac{\left|R_{0}-R_{g}\right|}{R_{0}}\times 100\%=\frac{\left|\Delta R\right|}{R_{0}}\times 100\%$$
where:$R_{0}$—sensor resistance without the presence of gas,$R_{g}$—sensor resistance in the presence of gas.

Another common definition of sensitivity is the dependencies [87,88]:(3)$$S=\frac{R_{0}}{R_{g}}-1$$

As for determining the responses of the sensor to gas, sensitivity uses not only resistance but also the conductance or current flowing through the sensor. The sensitivity of the sensor is expressed in per cent (2).

**Figure 11 sensors-22-02557-f011:**
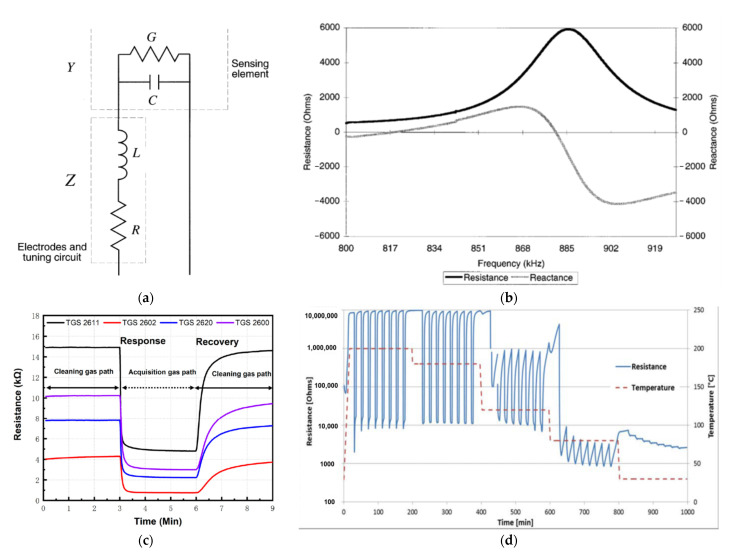
(**a**) Equivalent electrical circuit for the sensor. Reprinted with permission from [89] Copyright 2021 Elsevier; (**b**) Typical measured frequency response. Reprinted with permission from [89] Copyright 2021 Elsevier; (**c**) Process of each sampling: electromagnetic directional valve change gas path at 3 and 6 min. Reprinted with permission from [66]; (**d**) Resistance response of a Pd activated SnO_2_ thin-film semiconductor gas sensor for 1000 ppm (0.1%) H_2_ pulses in synthetic air at different temperatures. Note the high sensitivity, fast response and recovery at 180 °C and ability of low temperature (80–120 °C) operation. Reprinted with permission from [16] Copyright 2021 Elsevier.

In cases in which the gas-sensing characteristics are measured directly by measuring the resistance change of the gas-sensing material under exposure to target gas, different devices are used depending on the need. The order of magnitude that was measured during the measurements plays a decisive role here. In the case of semiconductor sensors, resistance, impedance, and current are most often measured. Depending on how big the measured values are and to what extent they change, a multimeter [17,69,82], digital multimeter modules [62], ohmmeter, electrometer, analyser [17,70,79], resistance bridge, picoammeter, and other devices are used. In electrical measurements of high resistances (characteristic of semiconductors), it is not only the measuring instruments that are important but also all connections to the measuring chamber, except for cables used for heating control, should be made using high-quality shielded Triax cables, and the shields should be earthed [76]. Due to the use of high-quality cables, electromagnetic interference has been eliminated. It is common practice to shield the test bench in order to minimise the effects of electromagnetic interference [21]. Another solution to minimise the impact of electromagnetic interference is the use of a resistor placed close to the sensor, and the use of a low-noise preamplifier is based on a low-noise-level low FET transistor in the input stage. This solution gives better results than the amplifier with bipolar transistors [21].

The meter for this type of measurement should be characterised by high accuracy in a very wide measuring range (from a few ohms to Gohms). The acquisition of the measurement data and the measurement conditions themselves takes place constantly over time; only in this way is it possible to know the exact reaction of the sensor to gas and to examine the mechanism of this response, shape etc. Amrani et al. [89] presented the use of alternating current (a.c.) to measure changes in gas-sensor impedance over a wide frequency range. This solution increased the performance by obtaining more useful information as a result of the measurements compared to the measurements of changes in electrical conductivity with the use of direct current (d.c.). Similarly, alternating voltage (a.c.) was used for measurements by Simons et al. [79], who used an impedance analyzer connected to a dielectric interface (SI 1260 and SI 1296; both Solartrons) to measure the electrical impedance, thus enabling measurements in the impedance range up to 1014 (±1%). In all measurements, the voltage was set to 0.1 V (rms) in order to stay in the linear regime [79]. The other measurement modes, such as voltage or capacitance changes, have been previously reviewed in the literature [21,64,90,91]. In 1997, Amrani et al. investigated the sensor’s response to the presence of gas with the use of alternating voltage (a.c.), determining the frequency characteristics of the tested sensors, and on their basis, they created an electric model of the gas sensor (Figure 11a,b) [89]. In 2007, Gurlo et al. published a review of the use of spectroscopy in measurements in situ and operando for assessing mechanisms of gas sensing [92]. An interesting example of an unusual approach to measuring the sensor response of gas sensors is the method presented by Nakata et al. [93]. This is based on the study of the dynamic nonlinear response of the sensor to gas using the fast Fourier transform. At the same time, in order to enhance the response of the semiconductor sensor to gas, cyclic temperature changes in the form of several harmonics were applied to it [93]. In 2019, Barauskas et al. used in situ measurement of simultaneous real-time resonance frequency detection and Fourier transform infrared spectroscopy in the investigated sensory response of methylated poly (ethylene) imine modified capacitive micromachined ultrasonic transducer [64,90].

Thus, the equipment used for the measurement of resistance changes is as important as other parameters. One of the common mistakes of resistance change measurements is measuring in different currents/voltages ranges; this can result in higher resistance changes under specific conditions and, in fact, a distorted sensor response or calculated sensitivity.

### 2.4. Static and Dynamic Measurements

Apart from the above-mentioned issues that have to be taken into account when gas sensors are tested, choosing the dynamic or static mode should be the first step.

#### 2.4.1. Static Measurements

In the static mode of measuring sensor properties, the reading of the value of changes in the magnitude of resistance (or other measured quantity, e.g., conductance, current, capacitance, etc.) is made in the steady state of the sensor or the quasi-steady state, and these constant values are used for further calculations and analyses. The results of such measurements are specific, determining values of resistance, current, etc., in the presence of gas and without the presence of gas. In these types of measurements, changes in the resistance value over time (their dynamics) are not interesting. Measurements of this type, depending on the tested samples, may last from a few minutes to many hours or even days. Due to their specificity, static measurements of the sensor response of gas sensors do not have such high requirements in terms of the measurement set-up. In this type of measurement, the volume of the measuring chamber and gas connections is of secondary importance. The stabilisation of temperature, humidity, etc., may take place at a slower pace due to the measurement time and measurement under steady-state conditions. Of course, the above-mentioned parameters, especially with regard to their stability, must not affect the measurement results. 

By contrast, in the dynamic mode of measuring sensor properties, changes in the measured quantity, most often resistance, are recorded over time [50,71,77,94]. Dynamic measurement, as the name suggests, progresses continuously—recording the dynamics of the sensor while feeding gas to the measuring chamber after stopping the gas dosing. As a result of such measurements, the characteristics of resistance [50] or sensitivity [76] over time are obtained, from which much more information can be drawn than from static measurements. Apart from the magnitude of changes in resistance, one obtains information about the rate of changes in resistance and their speed. It becomes possible to determine the response time and the recovery time of a given sensor to a specific gas concentration. Brief definitions of the aforementioned parameters describing the dynamics of the gas sensor’s behaviour and typical dynamic characteristics of the sensor’s response over time are presented in the literature [95] (Figure 12). In addition to gas concentration, other parameters can also be changed during measurements—such as humidity, temperature and flow—in order to study their influence on the behaviour of the sensor. With a sufficiently long measurement time with the dynamic method, the measured values will reach quasi-constant values, so the results obtained by the static measurement method are also possible to obtain using the dynamic method. In the dynamic method, apart from the measurement of the value of resistance (conductivity, current, capacitance, etc.), the time in which the changes of the measured value take place is also significant. Each delay in the system introduces a measurement error. Before starting the measurements, it is important to know what the delay in gas dosing is, i.e., how much time elapses from the opening of the gas dosing valve to the gas reaching the sensor in the measuring chamber. The volume of the gas connections and the measuring chamber itself is of decisive importance here.

#### 2.4.2. Dynamic Measurements

Kato et al. [71] presented the implementation of an intelligent gas detection system. In this work, the measuring station is presented in detail. The schematic diagram of the system to obtain a time-dependent nonlinear dynamic response is shown in Figure 13. Five different types of SnO_2_ gas sensors were tested. The sensors were checked for the following gases: air, ethanol, methanol, diethyl ether, acetone, ethylene, (salt) ammonia, isobutane and benzene. Each gas concentration is 100 ppm, except air. As a result of the measurements, phase diagrams for individual gases were obtained that showed characteristic loops. The data were processed using the time-dependent trace fast Fourier transform (FFT) method. On the basis of the obtained analyses, the researchers concluded that the characteristic data of the waveform contained information on current chemical species. In order to determine the gas concentration, a multiple regression model consisting of real data was used, as were imaginary harmonics. Eight regression equations corresponding to each classified gas species were obtained with a neural network and its concentration.

The authors concluded that using the combination of numerical computational procedures with the FFT neural network, and it can be suggested that regression has practical application as an “intelligent” gas sensor. The created system confirmed its reproducibility and its ability for classification and discrimination, and quantification.

In 2020, Kwon et al. [73] described a study in which gas was admitted to the measuring chamber with the tested sensor for only thirty microseconds, while the ventilation of the chamber lasted five seconds. Such short gas delivery times require precise valve control, small volumes in the system and high flows in relation to the connection volume and the measuring chamber.

Since the beginning of research on gas sensors based on metal oxides [96,97], one of the key issues has been the description of the mechanism responsible for gas detection [98,99,100]. Despite ongoing research in this area, many issues are still debatable. These include the emerging inconsistencies between electrical and spectroscopic studies and the lack of a proven mechanistic description of the surface reactions associated with gas detection. This is why the simultaneous measurement of the gas response with a determination of molecular adsorption properties is required for a better understanding of gas detection mechanisms [101]. These measurements are performed on clean and well-defined surfaces under appropriate conditions, i.e., ultra-high vacuum (UHV) or at pressures and temperatures simulating the actual “in vitro” sensor operating conditions [102].

The terms “in situ” and “operando” come from the quirky heterogeneous catalysis [103,104,105,106], in which the term “in situ” represents measurements for in situ catalyst testing, i.e., “under reaction conditions … or under conditions relevant to the reaction conditions” [104]. Operando, on the other hand, is a recently introduced term to describe the measurement techniques used to characterise a “working” catalyst. This method combines in situ measurements with the simultaneous monitoring of catalytic efficiency during the same experiment and on the same sample [105,107,108,109,110].

The measurement of the operando can be considered as “more perfect” [104] or “true” in situ experiments, in which the catalytic efficiency is measured simultaneously by the spectroscopic or structural properties of the experiment [106]. An example of an operando measurement diagram and photos of the measuring chamber enabling this type of measurement is shown in Figure 14.

The differences between in situ and operando measurements, as well as the need to distinguish between these terms, have been the subject of many lively discussions [105,107,108,109,110,111]. These terms are often used synonymously, as can be found in the literature [112], which highlights the different perceptions of these methods by researchers and requires an attempt to refine the definitions of these concepts. This task was undertaken by Gurlo and Riedel [92]. So far, no satisfactory consensus has been reached. In situ research methodology is still more widespread and is more common in research [103,104,106]. However, the operando method has been gaining popularity in recent years [105,107,108,111].

Figure 15 schematically shows the concepts of in situ and operando, while Figure 16 is a graphic summary of the work on in situ and operando methods in the period from 1985 to 2010.

Figure 17 shows the differences between an individual in situ and operando experiments. A comprehensive review of the application of Raman spectroscopy for working gas sensors with the use of in situ and operando methods was made by Elger A.K. and Hess Ch. in 2019 [113]. To summarise:In situ measurement is a characteristic of a material measured in operational conditions or conditions significant for operational conditions. The sensing properties of the tested material do not have to be characterised or are characterised in a separate experiment. For example, in 2001, Emiroglu et al. published a paper on in situ diffuse reflectance infrared spectroscopy study of CO adsorption on SnO_2_ [114]. In 2019, Barauskas et al. [90] performed an in situ experiment in which some of the measurements characterizing the properties of the gas-sensitive material were performed simultaneously with the measurement of sensor properties, and the rest of the research was carried out in a separate experiment [90]. Further examples of in situ experiments are presented in [20,64,70,74,79,115,116].Measurement operando is the real-time and operating characteristics of the active sensor element with simultaneous measurement of sensor properties and monitoring of the gas composition surrounding the sensor. In 2012, Sänze et al. [117,118] presented the measurement stand for an operando experiment. In 2014, Sänze et al. presented a detailed operand study of the indium gas ethanol gas detection mechanism by simultaneously measuring the sensor response (DC electrical conductivity), the Raman spectrum of the sensor material, and the Fourier transform infrared (FTIR) spectrum of the phase composition gas. This shows that detailed spectroscopic studies under operating conditions are necessary to explain the operation of gas sensors [119]. Degler et al. in 2015 reported that it is possible to track the surface chemistry of oxygen on SnO_2_-based gas-sensing materials using Fourier reflectance-transform operando spectroscopy (DRIFTS) [120], and the next year they wrote the first report of a successful measurement of UV/vis spectra recorded from an operating gas sensor [121]. Degler et al. in 2018 published another paper presenting the results of the measurements of the operando spectroscopy to unravel the complex structure—function-relationships which determine the gas sensing properties of Pt loaded SnO_2_ [122], and in 2019, they prepared a paper on operando research on the temperature-dependent interaction of water vapour with tin dioxide and its effect on gas detection [123]. Another example of an operando experiment on MOX gas sensors is the work of Elger et al. from 2019 [124]—they present the results of the combined operando Raman—gas-phase FTIR spectroscopy of ceria-based gas sensors during ethanol gas sensing. The authors use spectroscopy operando to deduce significant differences between the operating mode of gas sensors and catalysts [124]. In their second work, the authors presented the results of another experiment, which was titled *Elucidating the Mechanism of Working SnO_2_ Gas Sensors Using Combined Operando UV/Vis, Raman, and IR Spectroscopy* [125]. In 2019, a paper by Causer et al. was also published, presenting the results of the measurements of an operando investigation of the Hydrogen-Induced Switching of Magnetic Anisotropy at the Co/Pd Interface for Magnetic Hydrogen Gas Sensing [126].

The following boundary conditions of the operando experiment follow from the above definitions [92]:When treating the sensor as a whole, the sensor element is a complex device. It consists of many parts, for example, a semiconductor sensor with a measured electrical response that has a sensor layer deposited on a substrate on which in a row there are electrodes (for reading the electrical signal) connected to a transducer. Properly evaluating these interfaces is of utmost importance in understanding the detection mechanism.Real-time measurement. One of the basic assumptions when designing sensors is how to react quickly to changes in the gas atmosphere; therefore, a fast measurement response (e.g., spectroscopic) is essential.Testing under operating conditions that may vary significantly from ambient conditions (RT and atmospheric pressure) with regard to high temperatures and pressures.Simultaneous monitoring of the sensor activity, the output signal from the sensor (measured gas concentration) is transmitted by the sensor as an electrical signal or another type of signal depending on the type of sensor and the technology of the transducers used.Online parallel gas composition measurement is of great importance as it plays two roles. The output composition and concentrations provide information about the reaction products and its possible paths, while the input concentration enables the verification of the input data of the sensor (concentration of the detected gas).

**Figure 17 sensors-22-02557-f017:**
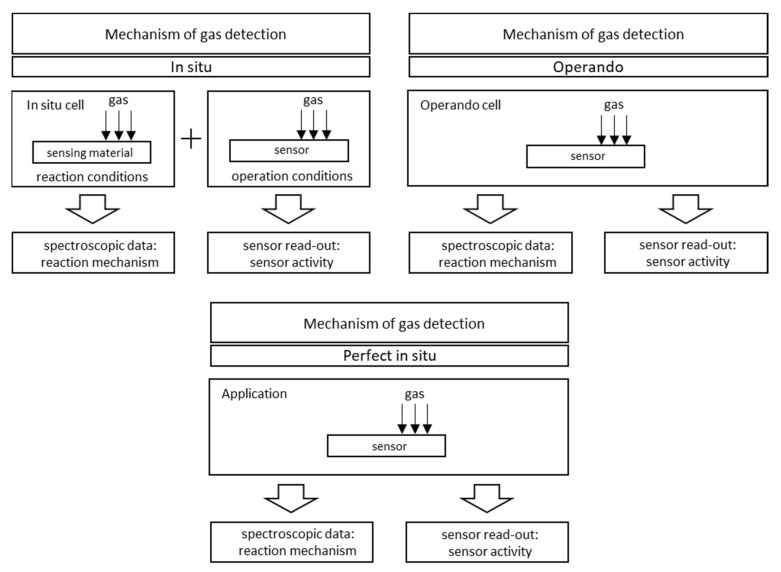
In situ and operando methodology in gas sensing.

In 2021, Valt et al. [127] presented the design and validation of a novel reaction chamber for operand Fourier spectroscopy in infrared with diffusion reflection for chemoresistive gas sensors, methodology of measurements and results. The measurement system can be helpful in the study of the gas–solid phase reaction on the surface of the sensor during its operation. The measuring chamber, the stand diagram and example sensor characteristics are shown in Figure 18.

## 3. Results and Discussion

Each research team usually constructs its own research set-up as far as technical and especially financial possibilities are concerned. This overview shows various test settings for the properties of gas-sensitive materials, including very simple set-ups that only allow basic measurements, e.g., sensor response or spectroscopy. In such set-ups, commonly available materials were often used, such as a syringe as a measuring chamber or as a dosimeter of a given gas volume—a multimeter was used as a measure of changes in sensor resistance. There are also specialised set-ups for static or dynamic response tests that differ significantly in principle. Set-ups are also built for testing many sensors at the same time, often with the possibility of changing the gas dosed to the measuring chamber.

Today, in the research of gas sensors, it is becoming standard to fully control and maintain the temperature, gas flow through the chamber and the humidity of the sensor operation with a wide range of regulations. Data are collected in real time. By using mass spectrometers at the entrance and exit to the measuring chamber, full control and knowledge about the reactions and processes taking place in the measuring chamber can be obtained.

For some time, a trend has been observed in the construction of measuring stations for in situ and operando measurements, used to measure sensors in their work environment. More and more often, there are measuring stations that allow the researcher to perform multiple measurements at the same time, which enables measuring the exact characteristics of the sensor during its operation.

Each measuring set-up during construction must be properly thought out and designed. The materials, equipment, gas connections, measuring chambers, mixers (especially their volumes), electrical connections, additional systems (maintaining and stabilising humidity and temperature, safe gas removal and neutralisation) may affect the results of sensor measurements—the researcher must be fully aware of these factors

## 4. Conclusions

The current research on gas-sensitive materials is not only focused on discovering new materials that change their properties in the presence of gas, at a specific concentration, temperature, humidity and for a specific period of time. More and more often, attempts are made to describe and characterise the reaction of a gas-sensitive material to gas and all processes taking place on the surface of a gas-sensitive material and in its volume. With these types of challenges, it is important to delve into research at the atomic level. Along with the increase in the detail of the measurements, the requirements for the measuring set-up increase. For static measurements, the volume of chambers and gas connections, flow, etc., are of no great importance. However, when measuring the dynamic properties of the sensor, the opposite is true; the test stand should be characterized by the shortest possible delays. The best solution is to map the actual working conditions of the sensor and perform all measurements at the same time, which guarantees that the results from different measurements are characterised by the same sample. Often, when performing individual measurements at different set-ups, during sample transfer and conditioning, many changes in the sample occur (changes in temperature, humidity, atmosphere, successive measurement cycles), which may cause significant changes in the properties of the sample, which may, in turn, lead to erroneous results and requests. Therefore, this review delivers an overview of various set-ups that were used in the gas sensor research over the last years.

The main goal of the review was to present different set-up used for gas-sensing analysis that has been applied over the last six decades of gas-sensing measurements. This tremendous task was performed to collect the data and analyse them. However, the components that were available 30–50 years ago do not enable the possibility to control the measurements in such good precision as it is performed now. There are many different types of test benches for testing sensor gas responses, each with advantages and disadvantages. It is important to use technology that is appropriate for the specific application. Based on the available knowledge and experience, power requirements, mounting requirements, and distance to other units must be considered. In addition, operating temperature ranges, gas calibration limits, accuracy and response times, and recovery times at different concentrations are important aspects. Moreover, there are several exciting application areas, for example, in the analysis of exhaled air biomarkers [128]. Future research should focus on multi-sensing using a gas sensor array [129], which is also of very good quality for ideal gas detection due to the high response and recovery times in gas detection applications. This solution provides and saves energy and power consumption, as well as being a small size. These advantages make it possible to develop a sensor that has the ability to detect more than one gas or an air pollutant. The most important aspect in the construction of test set-ups for sensors measurements is specialization. It is impossible to build a universal test set-up for gas sensors on a reasonable budget. The station should reflect the target work environment of the tested sensor. In scientific works, it is also important to include the characteristics of the test set-up in order to enable the correct interpretation of the obtained results by other scientists. Based on the literature review and the authors’ experiences, the test set-up has a large impact on the behaviour of the sensor. The sensor tested in different set-ups will give different results.

## Figures and Tables

**Figure 1 sensors-22-02557-f001:**
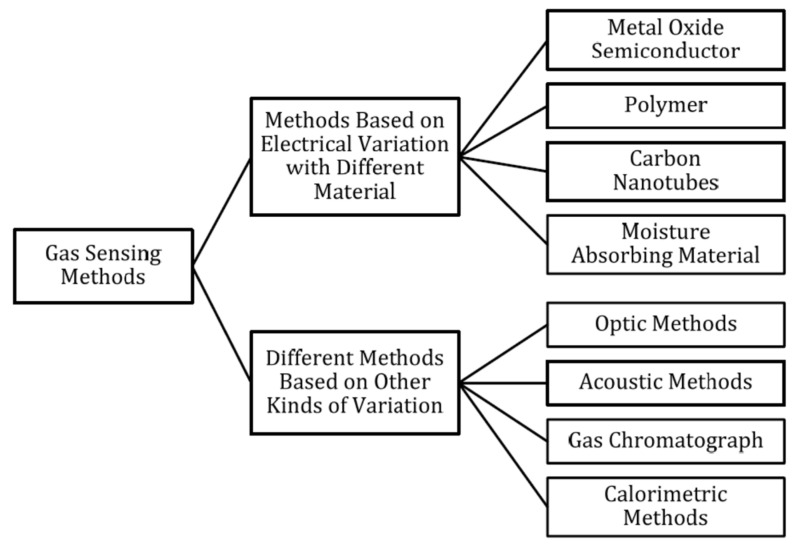
Classification of gas-sensing methods; reprinted with permission from [59].

**Figure 2 sensors-22-02557-f002:**
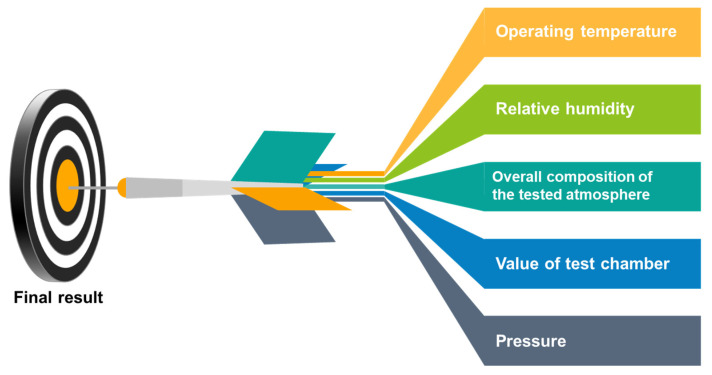
The main parameters that must be controlled in the stand for measuring the sensor response of semiconductor gas sensors.

**Figure 3 sensors-22-02557-f003:**
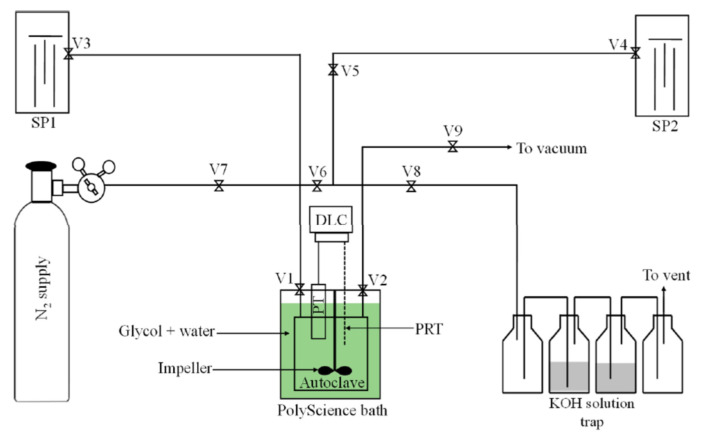
Schematic diagram of the set-up for the measurement of H_2_S hydrate dissociation in the presence of liquid water. V1, V2, DLC, and PT represent the inlet valve, outlet valve, data logging computer, and pressure transducer, respectively. V3, V4, V5, V6, V7, V8, and V9 represent control valves. SP1 is a high-pressure syringe pump containing degassed water, and SP2 is the high-pressure syringe pump containing H_2_S fluid. Reprinted with permission from [45].

**Figure 8 sensors-22-02557-f008:**
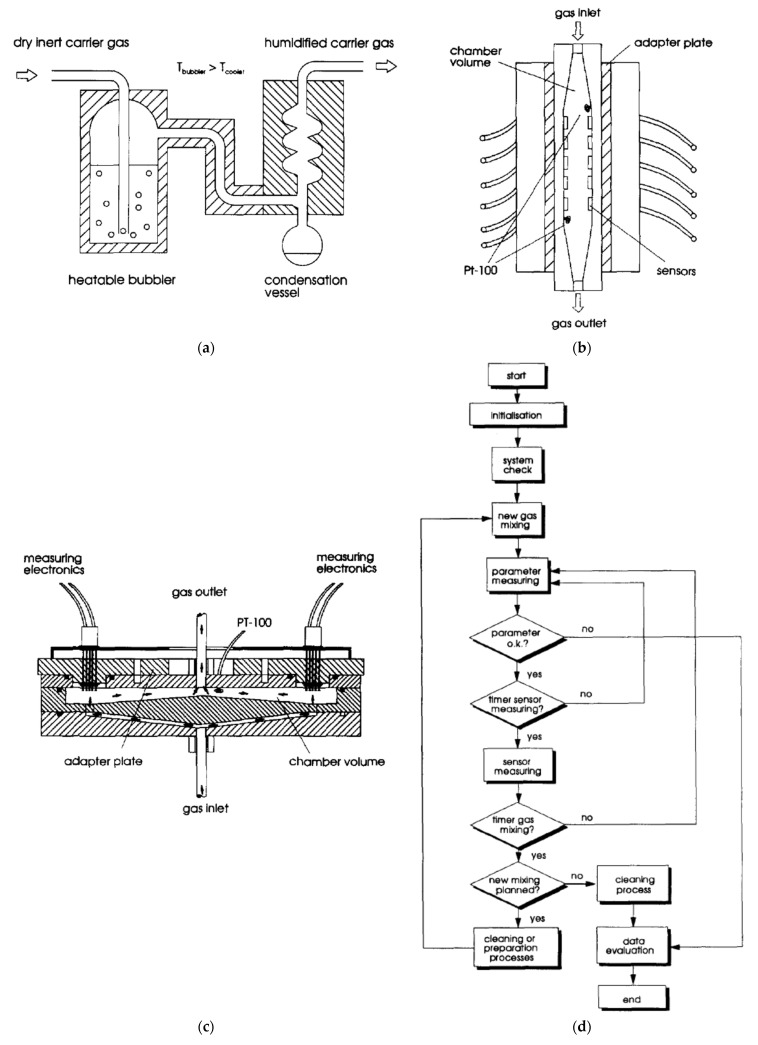
(**a**) Humidification set-up (bubbler and gas cooler). Reprinted with permission from [17]; (**b**) Plan of the linear measurement chamber. Reprinted with permission from [17]; (**c**) Plan of the circular measurement chamber. Reprinted with permission from [17]; (**d**) Schematic flow of the measurement cycle. Reprinted with permission from [17].

**Figure 12 sensors-22-02557-f012:**
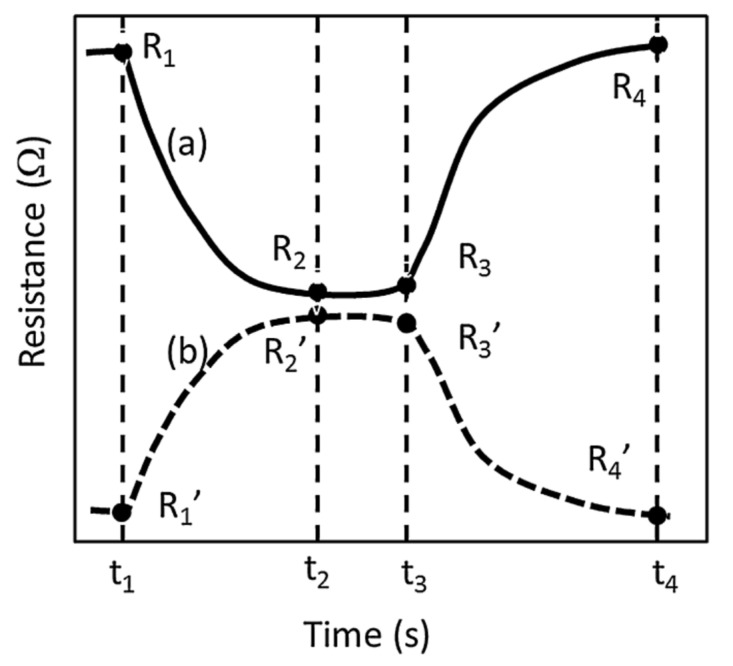
At time t_1_ an analyte gas is introduced; resistance of the sensor decreases in an oxidative (reductive) interaction until a critical time t_2_ when there exists no significant change. The measurement is continued until t_3_ to ensure completed diffusion or segregation of the analyte into the bulk or onto the surface of the sensor material. When the analyte gas is switched off at t_3_, the resistance starts to increase (decrease) until time t_4_. The resistances at the specified times are R_1_, R_2_, R_3_ and R_4_ in the oxidative curves (**a**) while they are primed in the reductive curves (**b**) Reprinted with permission from [95] Copyright 2021 Elsevier.

**Figure 13 sensors-22-02557-f013:**
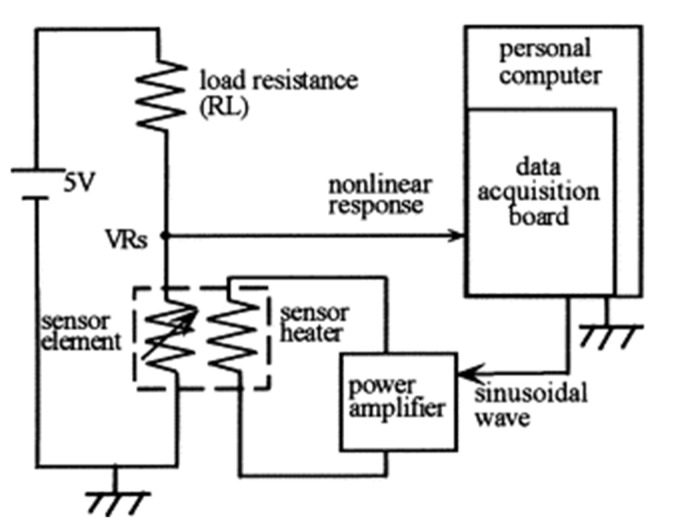
Experimental apparatus for detecting the dynamic response of a gas sensor. Reprinted with permission from [71].

**Figure 14 sensors-22-02557-f014:**
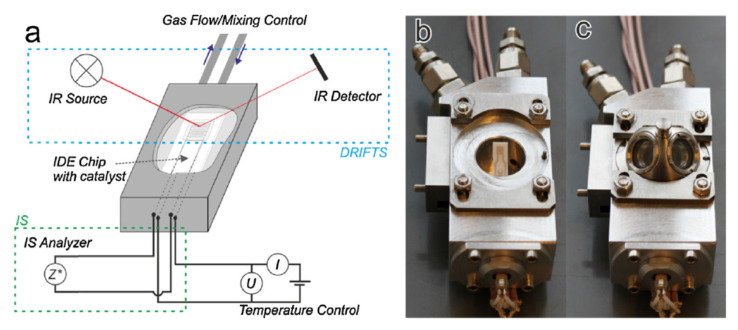
Measuring chamber for simultaneous IS and DRIFTS measurements with the same catalyst film. (**a**) Scheme of the chamber and measurement configuration; (**b**) photograph of open chamber; (**c**) photograph of chamber with dome. Reprinted with permission from [79].

**Figure 15 sensors-22-02557-f015:**
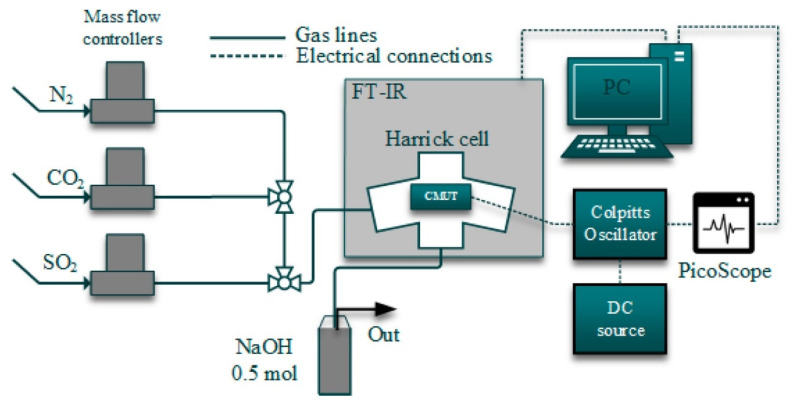
Experimental set-up for simultaneous real-time CMUT resonance frequency and the magnitude of electroacoustic impedance measurement and Fourier transform infrared spectroscopy. Reprinted with permission from [90].

**Figure 16 sensors-22-02557-f016:**
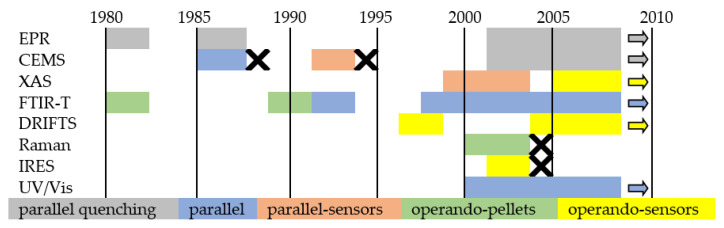
Development of in situ and operando techniques for gas-sensing studies on semiconducting metal oxides. X: no further works.

**Figure 18 sensors-22-02557-f018:**
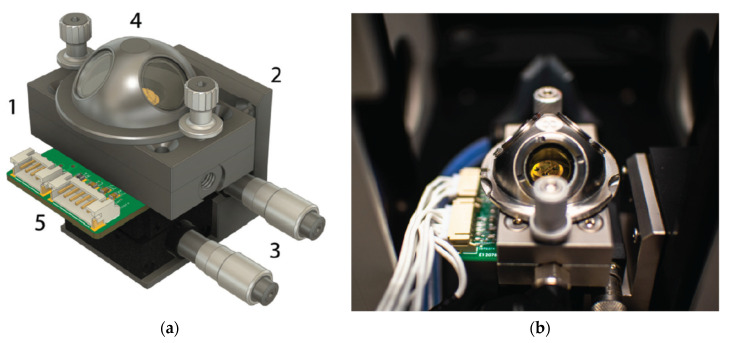
(**a**) 3D drawing and (**b**) photograph of operando test chamber with sensor mounted inside it. Reprinted with permission from [127].

## Data Availability

Not applicable.

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
