# Peer review of "A Review of Gas Measurement Set-Ups"

_sensors, 2022, doi:10.3390/s22072557_

Round 1

Reviewer 1 Report

This manuscript reviewed the various devices used in gas-sensing measurement over the last years, in addition, discussed and summarized the effects of different factors on the results of set-ups. This paper considered an important topic and was suitable for Sensors. Based on my reading, this manuscript could be published after a minor revision.

Some issues:

  1. Abbreviations in articles should be given full words when they first appear, such as MOS sensors (page 2, line 95), and MOX sensors (page 15, line 466).
  2. The order of the classification is confused, for example, Section2 appears twice, and Chapter 5 comes directly after Chapter 3.
  3. Section 2.1 is cumbersome and may need to be rescheduled.In my opinion, it would be better to list "Examples of gas-sensing set-ups" separately.
  4. According to the classification of "Dynamic and static measurements" in Section 2.2, the dynamic modes described in part I "Static measurements" should be rearranged to part II "Dynamic measurements". And then follow a series of examples to illustrate.

Author Response

Dear Reviewer,

Kind regards

Reviewer 2 Report

The review is elaborate and can be accepted for publication but needs some of the following corrections  to be added before proceeding for publication.

1. In the section  "2.1. Gas-sensing measurement set-up parameters" subsections should be numbered either by

2.1.1 Discharge and neutralisation of gases

2.1.2Materials used for the construction of measuring set-up 

or

i. Discharge and neutralisation of gases

ii. Materials used for the construction of measuring set-up 

similarly  other sections also should be numbered 

2. Authors discuss various set up involved in studying gas sensors. A table comprising of different methods and the % error in detection of gas using these methods need to be presented.

3. Authors can also make another table to discuss the advantages/disadvantages of each this technique in gas sensing 

4. Being a review article it expected  the data to be presented in form of tables  with referencing which is missing in the current review.

Author Response

Dear Reviewer,

Kind regards

Reviewer 3 Report

The growing demand for gas sensors leads to the increasing interest of scientists in this field. Up to now there are many papers on gas sensors composed of various materials and applying to lots of gases. Now Fuśnik et al. provided a comprehensive review on gas measurement set-ups. They have provided overall conclusions of how the different set-ups impact upon the obtained results. The equipment used for the measurement of resistance change in the gas measurement set-ups and Dynamic and static measurement methods have been introduced in detail. They pointed out that the factors such as materials, equipment, gas connections, measuring chambers, mixers, etc., which may affect the results of sensor measurements. The manuscript was well-written in formal English. The manuscript warrants publication after some proper revisions (some additions/edits) are made.

I have following concerns that the authors should address, clarify and include in the revised manuscript.

(1) Although the manuscript is a review on experimental development of various research techniques and various existing measurement systems, the theoretical role in developing gas sensoring materials is also non-negligible. Therefore, the authors should mention some typical theoretical exploration of nanomaterials for gas sensor applications. For instances, density functional theory has been utilized as an exploration tool for Tin Diselenide[D’Olimpio, G.; Farias, D.; Kuo, C.N.; Ottaviano, L.; Lue, C.S.; Boukhvalov, D. W.; Politano, A. Tin Diselenide (SnSe2) Van der Waals Semiconductor: Surface Chemical Reactivity, Ambient Stability, Chemical and Optical Sensors, Materials, 2022, 15, 1154], metal oxide nanoheets [Li, J.-H.; Wu, J.; Yu, Y.-X. DFT Exploration of Sensor Performances of Two-Dimensional WO3 to Ten Small Gases in Terms of Work Function and Band Gap Changes and I-V Responses, Appl. Surf. Sci., 2021, 546, 149104] and doped metal oxides [Pineda-Reyes, A.M.; Herrera-Rivera, M.R.; Rojas-Chavez, H.; Cruz-Martinez, H.; Medina, D.I. Recent Advances in ZnO-Based Carbon Monoxide Sensors: Role of Doping, Sensors, 2021, 21, 4425] as materials for gas sensors and this theoretical tool has discovered new materials that change their properties in the presence of gas and provided many material candidates for gas sensors. Adding two or more sentence to mention the theoretical role in exploration of gas measurement set-ups is essential for this review.

(2) Some essential comments on each gas measurement set-ups should be provided in order that the readers could easily understand their benefits and disadvantages.

(3) At the end of the manuscript, the authors should give the future-developing trends and prospective of the gas measuring set-up. In addition, some exciting application fields should also be pointed out. In this way, the manuscript would arouse great and wide interest of researchers.

(4) There are some minor errors or typos in the manuscript. For instances, <i> in Ref. 9, journal name “THE BELL SYSTEM TECHNICAL JOURNAL” should be corrected to “The Bell System Technical Journal”. <ii> The journal name of Ref. 19, 43, 48 and 49 are missing. <iii> Ref. 21 and 31, 39, 63, 68-70 are incomplete. <iv> Pages or article number of Ref. 26, 75 and 78 are missing. <v> The author names, year, volume and pages or article number in Ref. 80 are absent. <vi> Pages or article number in Ref. 87, 88, 90 and 91 are absent.

Author Response

Dear Reviewer,

Kind regards

Reviewer 4 Report

The manuscript entitled "A review of gas measurement set-ups" might be of  great interest for the young researchers working in the field.

Nevertheless, some shortcomings or weaknesses are present in the review, as follows:

  • in situ has to be in italics
  • Please, carefully check the references. It seems that they are not uniformly edited according to the rules.
  • The authors do not mention and do not cite the reports regarding the monitoring of gases at ambient conditions, that is the most actual important approach  (doi:10.5194/amt-5-1925-2012 or https://doi.org/10.3390/chemosensors8040134)  ......
  • The reprinted materials are too many and not originally integrated into the review, and often of very low quality (see Figure 7,  Figure 8).
  • The Results and discussion chapter does not bring the needed details and is not presented in a critical manner; it does not highlight the disadvantages/advantages of different approaches.
  • The review is too descriptive without outlining the most important issues to be solved.
  • The conclusions are more like perspectives.
  • A content should be introduced.
  • Some tables are important to clearly indicate the best functioning materials and the best design/construction systems.

Author Response

Dear Reviewer,

Kind regards

Round 2

Reviewer 2 Report

The authors have addressed  all the comments. The manuscript can be accepted for publication.

Reviewer 4 Report

The paper was improved and I agree to be published in its actual form.